# Efficacy of Transcranial Direct Current Stimulation (tDCS) on Cognition, Anxiety, and Mobility in Community-Dwelling Older Individuals: A Controlled Clinical Trial

**DOI:** 10.3390/brainsci13121614

**Published:** 2023-11-21

**Authors:** Nathalia Oliveira Rodrigues, Anna Alice Vidal Bravalhieri, Tatiane Pereira de Moraes, Jorge Aparecido Barros, Juliana Hotta Ansai, Gustavo Christofoletti

**Affiliations:** 1Institute of Health, Faculty of Medicine, Federal University of Mato Grosso do Sul (UFMS), Campo Grande 79060-900, Brazil; nathaliar861@gmail.com (N.O.R.); aabravalhieri@gmail.com (A.A.V.B.); tatianepm03@gmail.com (T.P.d.M.); 2Department of Physical Therapy, Dom Bosco Catholic University (UCDB), Campo Grande 79117-900, Brazil; jjorbarros@gmail.com; 3Department of Gerontology, Federal University of São Carlos (UFSCAR), São Carlos 13565-905, Brazil; jhansai@ufscar.br

**Keywords:** transcranial direct current stimulation, controlled clinical trial, aging, cognition, anxiety, gait

## Abstract

Transcranial direct current stimulation (tDCS) has gained popularity as a method of modulating cortical excitability in people with physical and mental disabilities. However, there is a lack of consensus on its effectiveness in older individuals. This study aimed to assess the efficacy of a 2-month tDCS program for improving physical and mental performance in community-dwelling older individuals. In this single-blinded, controlled clinical trial, forty-two participants were allocated to one of three groups: (1) the tDCS group, which received, twice a week, 20 min sessions of 2 mA electric current through electrodes placed on the dorsolateral prefrontal cortex; (2) the tDCS-placebo group, which underwent the same electrode placement as the tDCS group but without actual electric stimulation; and (3) the cognitive-control group, which completed crossword puzzles. Main outcome measures were cognition, mobility, and anxiety. Multivariate analyses of variance were employed. Significance was set at 5% (*p* < 0.05). Regarding the results, no significant benefits were observed in the tDCS group compared with the tDCS-placebo or cognitive-control groups for cognition (*p* = 0.557), mobility (*p* = 0.871), or anxiety (*p* = 0.356). Cognition exhibited positive oscillations during the assessments (main effect of time: *p* = 0.001). However, given that all groups showed similar variations in cognitive scores (main effect of group: *p* = 0.101; group × time effect: *p* = 0.557), it is more likely that the improvement reflects the learning response of the participants to the cognitive tests rather than the effect of tDCS. In conclusion, a 2-month tDCS program with two sessions per week appears to be ineffective in improving physical and mental performance in community-dwelling older individuals. Further studies are necessary to establish whether or not tDCS is effective in healthy older individuals.

## 1. Introduction

Aging is often associated with decline in cognitive and physical performance. As individuals age, they may experience a gradual reduction in mental processing speed, which can pose challenges in daily tasks that demand attention, concentration, and memory [1]. Additionally, significant changes occur in physical capacities, affecting the independence of older individuals [2,3]. To address the potential decline associated with aging, several therapies have been developed [4,5]. In recent years, a new technique called transcranial direct current stimulation (tDCS) has gained prominence by providing a constant and low-intensity direct current to specific areas of the brain.

tDCS is a noninvasive technique used to modulate neuronal activity and influence brain function. Previous studies have demonstrated encouraging outcomes with tDCS in various population groups, including young individuals and subjects with neurological disorders [6,7,8]. However, there is still uncertainty regarding the effectiveness of tDCS in older adults [9]. According to Guo et al. [10], the electric field generated by tDCS and its impact on cortical excitability depend on multiple factors, such as individuals’ age, lifestyle, tDCS montage, head diameter, and brain anatomy. Since older adults undergo age-related changes in brain structure and functionality, this aspect, which is challenging to control in clinical trials, may account for discrepancies in the benefits or lack thereof of tDCS.

The literature indicates that the short- and long-term effects of tDCS in older adults remain unclear. Current data show uncertainties around tDCS. According to Chase et al. [11], one reason for the difficulties in assessing tDCS findings is the potential of the brain to compensate through other networks. That is, many neural regions that are often targeted using tDCS have flexible coding properties and therefore may have the capacity to adapt to neural interventions [11,12].

Given the current state of research, caution is advised when drawing conclusions about the efficacy of tDCS in older individuals [13]. To address this gap, we conducted a comprehensive analysis of tDCS efficacy in community-dwelling older adults. Our study specifically tested the hypothesis that tDCS could serve as a valuable tool to improve both physical (mobility) and mental (cognition and anxiety) performance in comparison to a tDCS-placebo and to a control group.

## 2. Materials and Methods

This was a single-blinded controlled clinical trial with three parallel groups. This study was conducted at the Laboratory of Biomechanics and Clinical Neurology of the Federal University of Mato Grosso do Sul in the city of Campo Grande, Brazil. The study was conducted in accordance with the CONSORT statement, Declaration of Helsinki, and guidelines for good clinical practice. The protocol was approved by the Ethics Committee (#4.896.867, approval date 08.10.2021) and it was prospectively registered in the Brazilian Registry of Clinical Trials (#RBR-4nq8cbp). All eligible participants signed a consent form before the assessment.

The recruitment of subjects involved direct contact with potential participants and utilization of social media platforms. This process was extended across all districts of Campo Grande, Brazil to guarantee a comprehensive and impartial sample that accurately represented the city’s diverse population. The sample consisted of 42 individuals with a mean age of 71.6 years (standard deviation: 7.5), who were allocated to the experimental, placebo, or control groups.

Individuals were eligible for inclusion if they were 65 years or older, had no previous history of neurological or psychiatric disorders, were able to understand verbal commands, and comprehend the activities required for this research. Exclusion criteria included brain surgery, previous epileptic seizures, aneurysms, or other arteriovenous malformations. Additionally, the participants were screened for dementia. Individuals who exhibited scores on the Mini-Mental State Examination lower than the cutoff values adjusted for education and age established for the local population were excluded from the study. For this study, we used the following cutoff scores: 20 for illiterates; 25 for 1 to 4 years of schooling; 26.5 for 5 to 8 years of schooling; 28 for 9 to 11 years of schooling, and 29 for higher education levels [14,15].

### 2.1. Sample Size, Blinding, and Randomization

The sample size was determined using the G*Power software (version 3.1.9.7 for Windows, Heinrich-Heine-Universität Düsseldorf, Düsseldorf, Germany). The F test was employed with the following statistical parameters: (1) MANOVA, repeated measures, between factors; (2) computing the required sample size, given α, power, and effect size; (3) α error probability of 0.05; (4) power (1-β error probability) of 0.80; (5) three groups; (6) two measurements; and (7) an effect size of 0.42. The effect size was based on a study by Hsu et al. [16], who reported improvements in a noninvasive brain stimulation program in community-dwelling older individuals. The analysis revealed a noncentrality parameter of 11.642, F_critical_ value of 3.315, and minimum required sample size of 33 participants (11 in each group).

A total of 45 participants were enrolled in the study (15 in each group) to account for potential dropouts and to control for type I and II statistical errors. Figure 1 provides a detailed account of the number of participants who were included and excluded during the trial.

In this study, a single evaluator assessed all the participants. Regarding randomization, we initially proposed using randomly selected block sizes with sequentially numbered, opaque, and sealed envelopes to ensure equal distribution of participants among the three groups [17]. However, this trial was performed during the COVID-19 pandemic, and because of the recommendation for social isolation, some participants expressed concerns about attending in-person therapy sessions. Furthermore, despite international pharmaceutical laboratories offering vaccines, the Brazilian government postponed their purchase, leading to increased insecurity among participants. To prevent sample losses, the authors faced allocation limitations and decided to allocate participants by convenience. Participants who felt insecure about attending in-person sessions were assigned to the control group. The personal contact within this group was minimal.

The participants in the tDCS and tDCS-placebo groups were randomly assigned. While we recognized a bias caused by the convenient division of the control group, this approach was necessary to control for type I (α) and type II (β) statistical errors. It is important to note that the groups were similar in terms of sex, age, functional independence in performing activities of daily living (assessed with the Pfeffer Questionnaire, where higher scores reflect greater dependency) [18], cognition [14,19,20], anxiety [21], and mobility [22]. Table 1 provides details on the general characteristics of the participants.

### 2.2. Outcomes

Cognition was the primary outcome of this study. Anxiety and mobility were included as secondary outcomes. Assessments were performed at baseline and after eight weeks. All tests were performed in a randomized order to prevent potential order effects. The instruments used in this study were translated into Portuguese and validated for use in Brazilian participants.

Three instruments were used to assess cognition: the Mini-Mental State Examination (MMSE) [14], Frontal Assessment Battery (FAB) [19], and Semantic Verbal Fluency (SVF) test [20]. The MMSE evaluates participants’ overall cognition, including temporal and spatial orientation, word registration, attention and calculation, immediate and delayed recall, language, and visual-constructive skills. Scores ranged from 0 to 30 points, with higher values indicating better cognitive performance. FAB was used to quantify the executive function of the participants. The test provides insights into concept recognition, lexical flexibility, motor programming, conflicting instructions, inhibitory control, and environmental autonomy. The instrument scores range from 0 to 18 points, with higher scores indicating better cognitive performance. The SVF test was used to assess lexical knowledge and semantic memory organization. This test registered the number of animals that a person could count aloud for 60 s. A higher number of animals indicated better cognitive performance.

The Hospital Anxiety and Depression Scale (HADS) [21] was used to assess participants’ anxiety. This instrument includes 14 items, each scored from 0 to 3 for a total of 21 points. A score of 0–7 indicates an unlikely condition of anxiety; 8–11 suggests a possible condition of anxiety; and 12–21 indicates a probable condition of anxiety.

Mobility was evaluated using the timed get-up-and-go (TUG) test [22]. The results of this test included the time necessary to get up from a chair, walk 3 m, return, and sit down in the same chair. In this study, the TUG test was administered under three different conditions. The first condition followed the conventional administration method described by Podsiadlo and Richardson [22]. In the next two conditions, the TUG test was conducted with the inclusion of motor distractors, where participants carried a glass of water, and cognitive distractors, where they performed odd progressive number counting. This approach allowed for the comparison of TUG performance under standard conditions with situations that included additional motor and cognitive challenges.

### 2.3. tDCS Protocol

The intervention started 48 h after the baseline assessments. Participants in the experimental and placebo groups received, respectively, real tDCS (2 mA) and sham tDCS over the dorsolateral prefrontal cortex. Electrical stimulation was administered using a TCT tDCS stimulator device with a precision of ±0.004 mA, through a pair of rubber electrodes enclosed in saline-soaked sponge pockets. Each tDCS and tDCS-placebo intervention session lasted for 20 min. Both groups underwent two sessions per week for eight weeks.

In both the tDCS and the tDCS-placebo groups, tDCS was administered as a stand-alone intervention. In the tDCS group, the 2 mA condition started with a 30 s ramp-up to the desired intensity, which was maintained for 20 min before a 30 s ramp-down. In the tDCS-placebo group, the device automatically administered a 30 s ramp-up to 2 mA, followed immediately by a ramp-down to 0 mA. Electrode sizes were 25 cm^2^ (5 × 5 cm).

The active electrode (anode) was positioned over the left dorsolateral prefrontal cortex, according to the 10/20 International System for Electroencephalography. The returning (cathode) electrode was positioned over the supraorbital region. Electrode placements in the tDCS-placebo group were identical to those in the tDCS group. However, participants in this group received the current for 30 s and then slowly turned off without the participant’s knowledge. This has been shown to be a reliable and effective sham procedure to simulate sensations observed at the beginning of active stimulation without modifying cortical excitability [23].

The control group (receiving neither tDCS nor tDCS-placebo) received two crossword puzzles per week for eight weeks. Each puzzle corresponded to a session of the other groups. The puzzles had medium difficulty levels. The participants were instructed not to consult other people or external resources, such as the Internet or a dictionary, to assist in solving the puzzles. Unlike the other groups, which engaged in 20 min sessions, the control group did not have a specified time limit to complete the puzzles. In this study, errors in solving the puzzles were not computed.

### 2.4. Statistical Analysis

Statistical analysis was performed in several steps. First, we assessed the parametric assumptions of the data. Next, we performed multi- and univariate analyses of variance tests with Wilk’s Lambda and Greenhouse–Geisser test to examine the main effect of group, time, and group × time interaction. The effect sizes (*η*^2^*p*) were used in cases where statistically significant differences were observed. For all the analyses, significance was set at 5%.

## 3. Results

### 3.1. Efficacy of tDCS on Cognition

Table 2 displays the cognitive performance of the participants at baseline and final assessment. Multivariate analyses showed that the cognitive scores of the groups were similar (*p* = 0.101). There was a noteworthy upward trend in the scores over time (*p* = 0.001; *η*^2^*p* = 0.427). However, the absence of a significant group × time interaction (*p* = 0.557) suggests that the observed trends in cognitive performance were not influenced by the specific treatment received.

Univariate analyses confirmed that there were no significant differences in cognition between groups (MMSE, *p* = 0.067; FAB, *p* = 0.292; and SVF test, *p* = 0.266). There was a positive trend in the cognitive scores between the initial and final assessments (MMSE, *p* = 0.001, *η*^2^*p* = 0.347; FAB, *p* = 0.005, *η*^2^*p* = 0.186; and SVF test, *p* = 0.032, *η*^2^*p* = 0.112). However, no significant group × time interaction was observed (MMSE, *p* = 0.345; FAB, *p* = 0.504; and SVF test, *p* = 0.305). Figure 2 highlights the absence of significant group × time interactions on the cognitive tests.

### 3.2. Efficacy of tDCS on Anxiety

Table 3 provides a detailed overview of the participants’ anxiety. The analysis reveals that all groups demonstrated similar HADS scores, with no significant variations observed (*p* = 0.237). This suggests a consistent in anxiety across the diverse groups under investigation.

During the 2-month follow-up period, no significant changes in anxiety levels were detected among the participants (*p* = 0.355). Furthermore, the absence of a significant group × time interaction (*p* = 0.356) emphasizes that the observed patterns in anxiety scores were consistent across all groups throughout the entire study duration. Figure 3 shows the absence of significant group × time interaction.

### 3.3. Efficacy of tDCS on Mobility

Table 4 provides a comprehensive overview of the results from the TUG test at both the baseline and final assessments. The multivariate analyses revealed a consistent physical performance across all groups, irrespective of whether the test was administered conventionally or with the introduction of motor or cognitive distractions (*p* = 0.846). This suggests that the groups exhibited similar physical capabilities under various testing conditions. Over the 2-month treatment period, there were no significant variations in TUG test performance (*p* = 0.128), highlighting the stability of physical capabilities throughout the course of the interventions. Moreover, the absence of a significant group × time interaction (*p* = 0.871) indicates that the impact of the interventions on physical performance was consistent across all groups.

Further supporting these findings, univariate analyses confirmed the absence of significant differences between groups in any of the TUG test variants (conventional test, *p* = 0.854; TUG test with motor distractor, *p* = 0.670; and TUG test with cognitive distractor, *p* = 0.426). Additionally, no significant variations in TUG test performance were identified over the 2-month treatment duration for each test variant (conventional test, *p* = 0.458; TUG test with motor distractor, *p* = 0.155; and TUG test with cognitive distractor, *p* = 0.179).

Consistently, no significant group × time interactions were observed across the different TUG test conditions (conventional test, *p* = 0.469; TUG test with motor distractor, *p* = 0.802; and TUG test with cognitive distractor, *p* = 0.807). These comprehensive analyses reinforce the robustness and stability of physical performance across groups throughout the study period and under various interventions. Figure 4 shows the absence of significant group × time interaction for all TUG conditions.

## 4. Discussion

This study examined the efficacy of a 2-month tDCS program in a sample of 42 community-dwelling older individuals. The research design included both a sham-placebo and cognitive-control condition to investigate the impact of tDCS on physical and mental performance. Overall, the results suggest that tDCS may not be an effective technique for improving cognition, anxiety, and mobility in community-dwelling older adults, at least not over a 2-month treatment period. In this section, we compare the results of our study with those of other studies that have shown different outcomes. It is important for researchers, healthcare professionals, and the general public to understand the specifics of this study before drawing conclusions about tDCS.

We included cognition as the main outcome because it is a common concern among older individuals. Normal aging is associated with declines in specific cognitive abilities, such as processing speed, memory, language, and executive functions [1,24,25]. Recent advances in neuroscience have shed light on the underlying mechanisms of these cognitive changes, including reductions in gray and white matter volumes [1]. These changes are believed to contribute to the observed cognitive decline that is associated with aging. Therefore, it is crucial to explore strategies that can prevent or slow down cognitive decline to enhance the quality of life of older adults.

To stimulate cognition, we positioned tDCS electrodes in the dorsolateral prefrontal cortex. This area is located in the middle frontal gyrus and is involved in higher executive function, impulsive behaviors, attention, and working memory [26]. We expected that stimulating this area would improve the cognition of participants. However, the results did not confirm our hypothesis. A period of 20 min tDCS sessions, performed twice a week for 2 months, was not effective in improving cognitive scores compared to the tDCS-placebo and cognitive-control groups.

Several factors may explain the cognitive finding. First, it is possible that tDCS does not provide any benefits to cognition in older adults, although this seems unlikely given the positive outcomes seen in previous studies [27,28,29]. Second, a current of 2 mA may not be sufficient to improve cognition in community-dwelling older individuals. Recent studies used a current of up to 5 mA, suggesting that higher intensities may be more effective [29,30,31,32]. Third, it is possible that two weekly interval sessions are not long enough to provide cognitive benefits in healthy older individuals. For instance, Alonzo et al. [33] observed substantial effects of tDCS when applied daily. In fact, cohort studies with medium- and long-term follow-up suggest that it is unlikely that short periods of treatment could result in cognitive improvement unless there are underlying neurological diseases. Therefore, it is important to perform more studies to better understand the impact of tDCS on cognitive function in community-dwelling participants.

Furthermore, individuals in all groups achieved high MMSE scores at baseline, indicating limited room for improvement in cognitive scores on this instrument. This could explain why the benefits of noninvasive brain stimulation techniques such as tDCS are more frequently observed in subjects with cognitive impairment [32,34].

Despite the absence of a significant difference between groups, all participants displayed improved cognitive test scores at the final assessment. Statistical tests indicated significant improvement. However, whether these changes are clinically significant remains unclear. Given that benefits were observed across all groups, it is more plausible that the observed improvement stemmed from a learning response by the participants to the cognitive instruments, rather than the effect of tDCS. Further studies are necessary to delve deeper into this matter and to conduct more extensive investigations.

Anxiety was included in this study because the same area used for cognitive stimulation has been shown to be effective in improving neuropsychiatric symptoms [26]. However, unlike other studies [35,36], our findings did not show significant improvements in anxiety. We attribute this outcome to the deliberate selection of older individuals without a previous history of psychiatric disorders and who only exhibited “possible symptoms of anxiety”, as indicated by their HADS scores. By specifically choosing participants without psychiatric disorders, we aimed to isolate the effects of tDCS while minimizing the potential influence of preexisting mental conditions. tDCS may have a greater impact on anxiety in individuals with more pronounced symptoms at baseline. Further research is required to explore the potential effects of tDCS on anxiety in populations with varying levels of psychiatric symptoms.

Our study aimed to assess mobility under single- and dual-task conditions, as previous studies have demonstrated that performing a secondary task while walking can increase cognitive interference and the risk of falls [37,38]. The results did not show any benefits of tDCS. Whether the participants were simply walking or walking while performing a secondary task, no notable effects were observed with tDCS. We attribute the lack of mobility benefits to two specific factors. First, all participants demonstrated normal performance on the TUG test, indicating the absence of preexisting mobility issues. This suggests that a 2-month tDCS program may not be effective in improving mobility in community-dwelling older individuals without underlying mobility problems. Second, the dual-task conditions used in this study, such as walking while carrying a glass of water or walking while performing progressive counting, may not have been sufficiently challenging for individuals with preserved cognitive and physical functions. Incorporating more demanding or challenging tasks in future assessments could potentially yield different outcomes in the tDCS group. Recent studies have adopted daily dual tasks, such as walking while talking on the phone or texting messages, to evaluate the cost associated with dual-task performance [39,40]. We encourage new studies to explore more challenging tasks in older adults with different cognitive skills, as this may provide further insights into the effects of tDCS on mobility and cognitive-motor interference.

In addition, we employed only one task to evaluate the effect of tDCS on motor function. This can be perceived as a weakness given that the electrodes were placed on the dorsolateral prefrontal cortex rather than on the primary motor cortex, which is the central region for motor control. We encourage further research to address this limitation.

In this study we observed a 6.7% sample size loss. It is important to note that none of the dropouts were associated with tDCS intervention; rather, they were a result of circumstances related to the COVID-19 pandemic. In addition, nonadherence to the tDCS protocol was not linked to the mental or physical prognosis of the participants. Given that dropouts occurred for reasons unrelated to the treatment being studied, we did not run either the intention-to-treat (ITT) principle or per-protocol set analyses [41].

### Limitations

Our findings should be interpreted in light of the following limitations. First, the results were restricted to community-dwelling older individuals without any neurological, psychiatric, or walking disability. Second, the control group was not randomly allocated. This introduces the possibility of a hidden effect that was not measured in this study. Third, while participants were blinded to the intervention (particularly in the case of the tDCS and tDCS-placebo groups), the assessor was not completely blinded and was aware of the group to which each participant belonged. Fourth, this study was conducted over a relatively short period, making it challenging to observe the full extent of the changes resulting from the interventions. Future large-scale studies should include 6- or 12-month follow-ups to more accurately evaluate the efficacy of tDCS in healthy older individuals.

## 5. Conclusions

In this study, we found that 20 min sessions of 2 mA conducted twice a week over a 2-month treatment period did not lead to significant improvements in physical (mobility) and mental (cognition and anxiety) performance in community-dwelling older individuals. Continued investigation is important to gain a more comprehensive understanding of the potential benefits and limitations of tDCS in healthy older adults.

## Figures and Tables

**Figure 1 brainsci-13-01614-f001:**
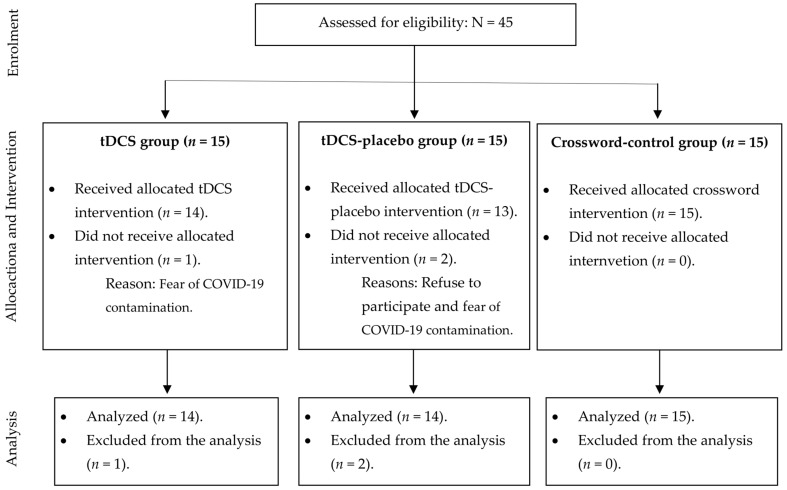
Description of number of participants included in each group.

**Figure 2 brainsci-13-01614-f002:**
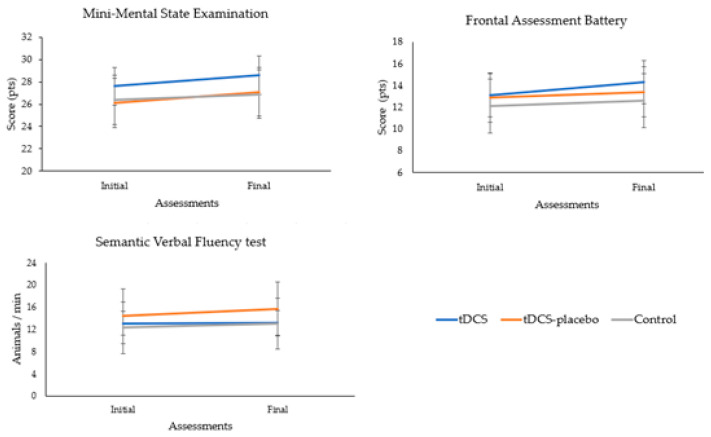
Analysis of the group × time interactions of the cognitive tests.

**Figure 3 brainsci-13-01614-f003:**
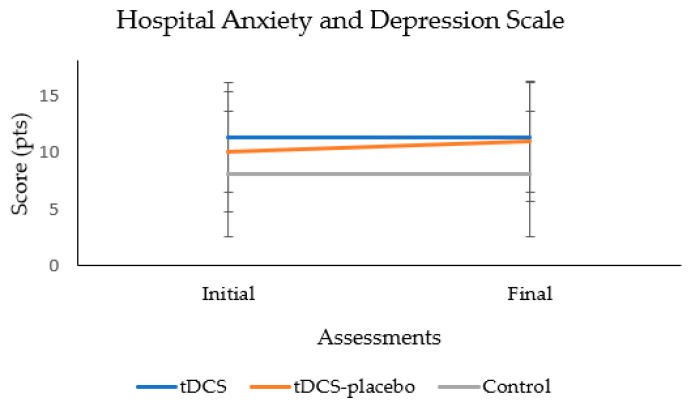
Analysis of the group × time interaction of the anxiety test.

**Figure 4 brainsci-13-01614-f004:**
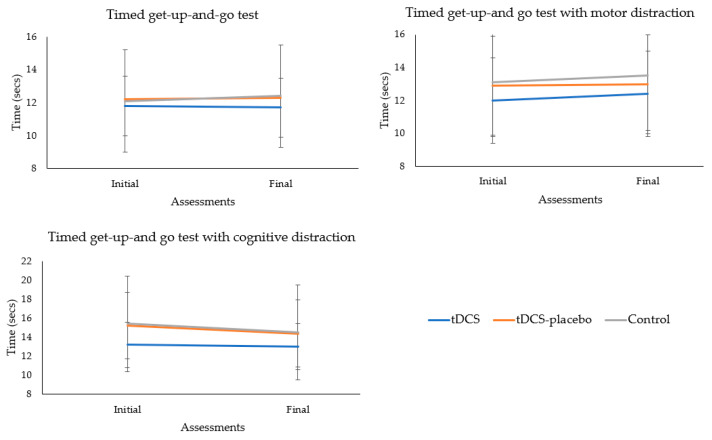
Analysis of the group × time interactions of the mobility tests.

**Table 1 brainsci-13-01614-t001:** General characteristics of the participants.

Variables	Groups			*p*
	tDCS	tDCS-Placebo	Control	
Sex (men:women), n	6:8	3:10	5:10	0.552
Age, years	71.3 (7.6)	70.9 (7.6)	72.6 (7.8)	0.830
Pfeffer Questionnaire, pts	0.4 (0.8)	0.5 (0.9)	0.3 (0.8)	0.815
Mini-Mental State Examination, pts	27.6 (1.7)	26.1 (2.2)	26.4 (1.9)	0.126
Frontal Assessment Battery, pts	13.1 (2.0)	12.9 (2.3)	12.1 (2.7)	0.510
Semantic Verbal fluency, animals/minute	13.1 (2.1)	14.4 (4.6)	12.3 (4.9)	0.401
Hospital Anxiety and Depression Scale, pts	11.3 (4.7)	10.0 (5.1)	8.1 (5.2)	0.246
Timed Get Up and Go test, secs	11.8 (1.7)	12.2 (3.2)	12.1 (3.2)	0.938

**Table 2 brainsci-13-01614-t002:** Cognitive performance of the groups at the initial and final assessment.

Cognition	Groups						MANOVA Main Effect	
	tDCS		tDCS-Placebo		Control			
	Assessment		Assessment		Assessment			
	Initial	Final	Initial	Final	Initial	Final		
Mini-Mental State Examination, pts	27.6 (1.7)	28.6 (1.1)	26.1 (2.2)	27.1 (2.0)	26.4 (1.9)	26.9 (2.2)	Group	*p* = 0.101
Frontal Assessment Battery, pts	13.1 (2.0)	14.3 (2.0)	12.9 (2.3)	13.4 (2.8)	12.1 (2.7)	12.6 (2.5)	Time	*p* = 0.001
Semantic Verbal Fluency test, animals/min.	13.1 (2.1)	13.2 (2.2)	14.4 (4.6)	15.7 (4.6)	12.3 (4.9)	13.0 (4.9)	Interaction	*p* = 0.557

Data are expressed in mean (standard deviation). Repeated-measures analysis of variance tests revealed a significant main effect of time in the MMSE, FAB, and SVF tests (*p* < 0.05). However, no significant main effect was observed for group or for group × time interaction (*p* > 0.05).

**Table 3 brainsci-13-01614-t003:** Anxiety of the groups at the initial and final assessment.

Anxiety	Groups						ANOVA Main Effect	
	tDCS		tDCS-Placebo		Control			
	Assessment		Assessment		Assessment			
	Initial	Final	Initial	Final	Initial	Final		
Hospital Anxiety and Depression Scale, pts	11.3 (4.7)	11.3 (4.8)	10.0 (5.1)	10.9 (5.4)	8.1 (5.2)	8.1 (6.0)	Group	*p* = 0.237
							Time	*p* = 0.355
							Interaction	*p* = 0.356

Data are expressed in mean (standard deviation). Repeated-measures analysis of variance tests revealed no significant main effect for time, for group, or for group × time interaction (*p* > 0.05).

**Table 4 brainsci-13-01614-t004:** Mobility scores of the participants at the initial and final assessment.

Mobility	Groups						MANOVA Main Effect	
	tDCS		tDCS-Placebo		Control			
	Assessment		Assessment		Assessment			
	Initial	Final	Initial	Final	Initial	Final		
Timed get-up-and-go test, secs	11.8 (1.7)	11.7 (1.8)	12.2 (3.2)	12.3 (2.8)	12.1 (3.2)	12.4 (3.5)	Group	*p* = 0.846
Timed get-up-and-go test with motor distraction, secs	12.0 (2.6)	12.4 (2.6)	12.9 (3.9)	13.0 (4.1)	13.1 (3.8)	13.5 (2.9)	Time	*p* = 0.128
Timed get-up-and-go test with cognitive distraction, secs	13.2 (2.4)	13.0 (2.4)	15.2 (3.7)	14.4 (3.3)	15.4 (6.8)	14.5 (5.6)	Interaction	*p* = 0.871

Data are expressed as mean (standard deviation). Repeated-measures analysis of variance tests revealed no significant main effect for time, for group, or for group × time interaction on the Timed get-up-and-go test with and without dual-task distractors (*p* > 0.05).

## Data Availability

The data presented in this study are available on request from the corresponding author.

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
