# Peer review of "Efficacy of Transcranial Direct Current Stimulation (tDCS) on Cognition, Anxiety, and Mobility in Community-Dwelling Older Individuals: A Controlled Clinical Trial"

_brainsci, 2023, doi:10.3390/brainsci13121614_

Round 1
Reviewer 1 Report
Comments and Suggestions for Authors
Introduction
Please provide more information about the current controversies and your solution to address these controversies. Besides, the novelty of the study in not clear.
Method:
1. What do you mean by “private residence”?
2. Please report the exact value of MMSE cut off score.
3. What is “Pfeffer index”?
4. Please provide more information in Fig1. (e.g. intervention in each group, the measurement points, )
5. Considering the blindness of assessor and assigning the participants in three different groups, why your study in “single-blinded?”
6. Have you used the Brazilian version of Frontal Assessment Battery and Hospital Anxiety and Depression Scale?
7. Is 2 sessions per week enough to improve the brain function? How many days was between each session of treatment?
8. How long was the ramp up/ramp down time?
9. tDCS in tDCS group and tDCS-placebo group was used as a stand-alone intervention?
Reviewer 2 Report
Comments and Suggestions for Authors
Thank you for giving me the opportunity to review this manuscript.
This study revealed that tDCS was not effective in cognition, mood, or morbidity in elderly populations.
1) Please describe clearly that tDCS was not effective in the title.
2) In order to demonstrate the effects of tDCS, I think that it is unnecessary to compare it with the control group, because the control was not randomized, the control did not have any sham effects (placebo effects) of tDCS, and so on. I think that the results of the control were not important at all, so it is better to delete the sentences of "Cognition showed positive oscillations during assessments (P = 0.001). However, as this trend was 28 observed in all groups, it is more likely that it was a learning response of the participants to the 29 cognitive instruments, rather than an effect of tDCS." in the abstract.
3) Please describe more clearly the methods of random sequence generation, and the allocation concealment betwen the active and sham tDCS groups.
4) Please describe whether ITT analyses or per-protocol set analyses were performed.
5) Please describe clearly the outcome measures and time point of the primary outcome.
6) Please attach the CONSORT 2010 checklist and fill in the page numbers.
I think it is necessary to revise the manuscript.

Reviewer 3 Report
Comments and Suggestions for Authors
The general purpose of the study was to examine the influence of a two month tDCS intervention on physical and mental performance in healthy older adults. A total of 42 subjects completed the study. These subjects were allocated to a tDCS group, a tDCS placebo group, and a crossword control group. The study was single blinded. The tDCS intervention involved anodal stimulation of the left DLPFC with an intensity of 2 mA for 20 minutes per session. The tDCS or placebo stimulation was applied twice a week for 8 weeks. The primary outcome variables involved cognitive measures since tDCS was applied to DLPFC. These included the MMSE, he FAB, and the SVF. In addition, the HADS was used to assess anxiety. The one motor test was the common lower body assessment of the Timed get up and go (TUG) test, although 3 variations of it were conducted.
The main findings were:
1) there were no significant differences in any of the 3 cognitive measures between groups, any improvements seen pre to post were similar between groups; 2) there were no significant differences in the HADS between groups; 3) there were no significant differences between groups for the TUG tests. In summary, tDCS did not enhance cognitive and motor performance in the current task conditions.
The study had several major strengths: 1) It was a long-duration study with a standard set of tDCS parameters (montage, duration, current levels etc); 2) The 2 control groups were a strong design feature along with the sample size estimation. Good overall methodology.3) Writing was good overall.
Overall, I liked this study. It appeared to be conducted carefully, was easy to understand, and most of the methodology/design used is standard in the field. I think the study adds to the literature on the topics involved. The study should be of interest to readers of Brain Sciences and researchers is several related fields. I think the researchers should be commended for publishing non-significant results. Oftentimes non-significant results are unfairly evaluated by some journals/reviewers, which really damages research in this area as it leads to losses of research resources/unnecessary replication, etc etc. Sometimes negative results provide just as much useful information as positive results.
The study did have a few weaknesses that must be addressed however.
1) The study was single blinded. This is an issue as nowadays most tDCS studies should be double blinded which has been emphasized in some review articles. However, the results were negative so I personally doubt there was any bias. In addition, the rater was blinded and it had 2 control groups. Please add single blinded as a limitation in the Limitations section.
2) The study was not randomized but there was a good reason for this (Covid etc).
3) The use of only 1 motor task (although 3 variants) could be viewed as a weakness. Relatedly, it was a lower body task and lower body muscles may not be as susceptible to tDCS especially when it is not applied directly to the M1 representation area of the muscles. This issue may also need to be added to the Discussion, which needs more content anyway (see below). Electrode size needs to be mentioned in Methods for tDCS.
4) The stimulation of only being 2 days a week and not multiple days in a row in a week (e.g. 4 or 5 consecutive days). This is perhaps the biggest issue with the study and the one that tDCS advocates would point out. The authors should point this out as a limitation in the Limitation section. Also they should probably add some text to the Discussion on this topic (Discussion needs improvement anyway) and cite some of the studies on the topic https://pubmed.ncbi.nlm.nih.gov/22037139/ https://pubmed.ncbi.nlm.nih.gov/37041183/ are two example studies that I am aware of. There are a few more every other day studies vs consecutive day studies I believe but I could not locate them. The authors should try to find these and add them to the Discussion with the above.
5) As alluded to above the Discussion needs to be improved. In the most basic sense there needs to be much more integration of their results with the current literature involving similar long-term tDCS studies in the cognitive and motor domains in older adults (not as much literature exists) and in older populations with motor disorders (more long term tDCS studies exist here). This is because this study involved long term tDCS with both motor and cognitive features. Thus, the Discussion needs integration among all these issues. At present the authors mainly just go over their results without any overall literature context. If anything they could use this literature to strengthen their findings as there have also been long-term tDCS cognitive and motor studies in old adults and older motor disorder populations that have shown no effect. This will improve the paper substantially.
This is the major reason it was stated above that the Discussion needs expansion and improvement. Thus, the above interrelated topics need to be covered. A few basic searches yielded several relevant long term tDCS motor and cognitive studies by different research groups in older adults and other populations that could be used. The authors could use these and I am sure they could find some additional ones to accomplish the above. Both positive and negative studies should be mentioned.
6) Overall, the paper was well-written and has few grammatical or typographical errors. However, there are some very minor English, formatting, and other errors. Another proofreading is needed here are a few of thee minor errors.
a. Introduction has short, choppy paragraphs (3 of them have only 2 sentences each) a few could be combined. The Introduction also needs more a little more bulk and background on the various topics, it is kind of short and too basic.
b. Line 192 needs a blank line after it.
c. Lines 26 and 198 the 5% should read P < 0.05 right?
d. Line 28 probably needs reworded I don’t think the reader is going to understand what that means in the abstract alone at all without reading the whole paper.
7) There are a few errors in the bibliography where some titles of articles have all the words capitalized whereas others do not. For instance, compare references 2, 21, 26, and 27 to most of the other references in the bibliography as just some examples.
8) Authors should consider using figures in addition to the tables to display the results. Other than the CONSORT the study has no figures at all.
Comments on the Quality of English LanguageMinor english editing and formatting.
Round 2
Reviewer 1 Report
Comments and Suggestions for Authors
There is no new comment
Reviewer 2 Report
Comments and Suggestions for Authors
I think this manuscript would be suitable for publication in this journal.
Reviewer 3 Report
Comments and Suggestions for Authors
The authors have been very responsive to my prior review and have made almost all of the changes suggested. The paper is substantially improved. My only remaining comment is that it would be good to see data figures. In addition, minor proofreading.
Comments on the Quality of English Languageminor English issues some proofreading
